# Measures of malaria transmission, infection, and disease in an area bordering two districts with and without sustained indoor residual spraying of insecticide in Uganda

Joaniter I. Nankabirwa[1,2]*, Teun Bousema[3,4], Sara Lynn Blanken[3], John Rek[1], Emmanuel Arinaitwe[1], Bryan Greenhouse[5], Philip J. Rosenthal[5], Moses R. Kamya[1,2], Sarah G. Staedke[6], Grant Dorsey[5]

1 Infectious Diseases Research Collaboration, Kampala, Uganda, 2 Department of Medicine, Makerere University College of Health Sciences, Kampala, Uganda, 3 Department of Medical Microbiology, Radboud University Nijmegen Medical Centre, Nijmegen, The Netherlands, 4 Department of Immunology and Infection, London School of Hygiene and Tropical Medicine, London, United Kingdom, 5 Department of Medicine, University of California San Francisco, San Francisco, CA, United States of America, 6 Department of Infectious Diseases, London School of Hygiene and Tropical Medicine, London, United Kingdom

* jnankabirwa@yahoo.co.uk.

## Abstract

Tororo District, in Eastern Uganda, experienced a dramatic decline in malaria burden starting in 2014 following the implementation of indoor residual spraying of insecticide (IRS) in the setting of repeated long-lasting insecticide treated nets (LLINs) distribution campaigns. However, in 2020 malaria began to resurge in Tororo following a change in the active ingredient used for IRS. In this study, epidemiological measures of malaria were compared shortly after the resurgence between two parishes in Tororo District (Kayoro and Osukuru) and one contiguous parish in Busia District (Buteba), where IRS has never been implemented. A cohort of 483 residents from 80 randomly selected households were followed from August 2020 to January 2021. Mosquitoes were collected every 2 weeks using CDC light traps in rooms where participants slept; parasitemia and gametoctyemia measured every 4 weeks by microscopy and PCR; and symptomatic malaria measured by passive surveillance. The annual entomological inoculation rate was significantly higher in Buteba (108.2 infective bites/person/year), compared to Osukuru (59.0, p = 0.001) and Kayoro (27.4, p<0.001). Overall, parasite prevalence was 19.5% by microscopy and 50.7% by PCR, with no significant differences between the three parishes. Among infected individuals, gametocyte prevalence by PCR was 45.5% and similar between sites. The incidence of malaria was significantly higher in Osukuru (2.46 episodes PPY) compared to Buteba (1.47, p = 0.005) and Kayoro (1.09, p<0.001). For participants over 15 years of age, the risk of symptomatic malaria if microscopic parasitemia was present was higher in Osukuru (relative risk [RR] = 2.99, p = 0.03) compared to Buteba. These findings highlight the complex relationships between measures of malaria transmission, infection, and disease, and the potential for excess disease burden,

**Data Availability Statement:** Data from both cohort studies are available through a novel open-access clinical epidemiology database resource, ClinEpiDB. https://clinepidb.org/ce/app.

**Funding:** GD, MK, Research: Funded by the National Institutes of Health as part of the International Centers of Excellence in Malaria Research (ICMER) program (U19AI089674) JIN supported by the Fogarty International Center (Emerging Global Leader Award grant number K43TW010365) EA is supported by the Malaria Training Program supported by the Fogarty International Center of the National Institutes of Health under Award Number D43TW010526. TB and SLB are supported by a fellowship from the European Research Council (ERC-CoG 864180; QUANTUM). The funders had no role in the study design, data collection and analysis, decision to publish or preparation of the manuscript.

**Competing interests:** The authors have declared that no competing interests exist.

possibly due to waning immunity, in areas where vector control interventions begin to fail after a sustained period of highly effective control.

## Introduction

Significant progress in malaria control has been realized in the last decade in Uganda [1]. This has been attributed to the scale up of effective control interventions including case management with artemisinin-based combination therapies, mass distribution of long-lasting insecticide treated nets (LLINs), intermittent preventive treatment in pregnancy (IPTp), and re-introduction of indoor residual spraying (IRS) in some districts. Together, these interventions have resulted in declines in measures of transmission intensity and parasite prevalence in children under 5 years of age [1–3], although this progress has not been unform across the country and has stalled in the last few years [4].

One dramatic example of progress is Nagongera sub-county, located in the central part of Tororo District in Eastern Uganda, a historically high transmission area. Comparing key malaria indicators measured from 2011–14, prior to IRS, and in 2017–19 after two national mass LLIN distribution campaigns and 5 years of sustained IRS, dramatic reductions in estimates were observed, with the entomological inoculation rate (EIR) falling from 238 to 0.43 infectious bites/person/year, prevalence of parasitemia measured by microscopy in children aged 0.5–10 years from 32% to 6.8%, and incidence of malaria from 3 to 0.05 episodes/person/year [2]. Although this dramatic decline in malaria burden following intense and sustained vector control suggests that elimination may be feasible in a previously hyperendemic area of Uganda, more recent data highlight the fragile nature of these gains. Beginning in 2020, a marked resurgence in malaria cases was documented in 5 districts of Uganda (including Tororo) following a change in IRS formulation, reaching pre-IRS levels within 1–2 years [5]. Although the causes of this resurgence are unclear, it is important to better understand the underlying epidemiology of malaria in areas where resurgence has occurred. In this study, we compared measures of malaria transmission, infection, and disease in a contiguous border area that included two parishes in Tororo District where control effects began to fail after 6 years of sustained IRS and one parish in Busia District, where IRS has never been implemented. Our underlying hypothesis was that there would be important difference in epidemiological measures of malaria between these two regions that could further our understanding of the epidemiological consequences of resurgent malaria in area where vector control interventions have diminished effectiveness.

## Methods

### Study design and setting

Between August 2020 and January 2021, a comprehensive cohort study and entomological surveillance were conducted in a contiguous area of Eastern Uganda that included two parishes in Tororo District (Osukuru and Kayoro parishes) and one parish in Busia District (Buteba parish) (Fig 1). To provide contemporaneous data from the interior of Tororo District (Nagongera Health Centre) and within the area of Tororo District where the cohort study was conducted (Osukuru Health Centre), routine health facility-based data on malaria morbidity was collected from January 2015 through June 2021.

Before 2013, malaria control in Tororo District was limited to the distribution of LLINs through antenatal care services, promotion of intermittent preventive treatment during

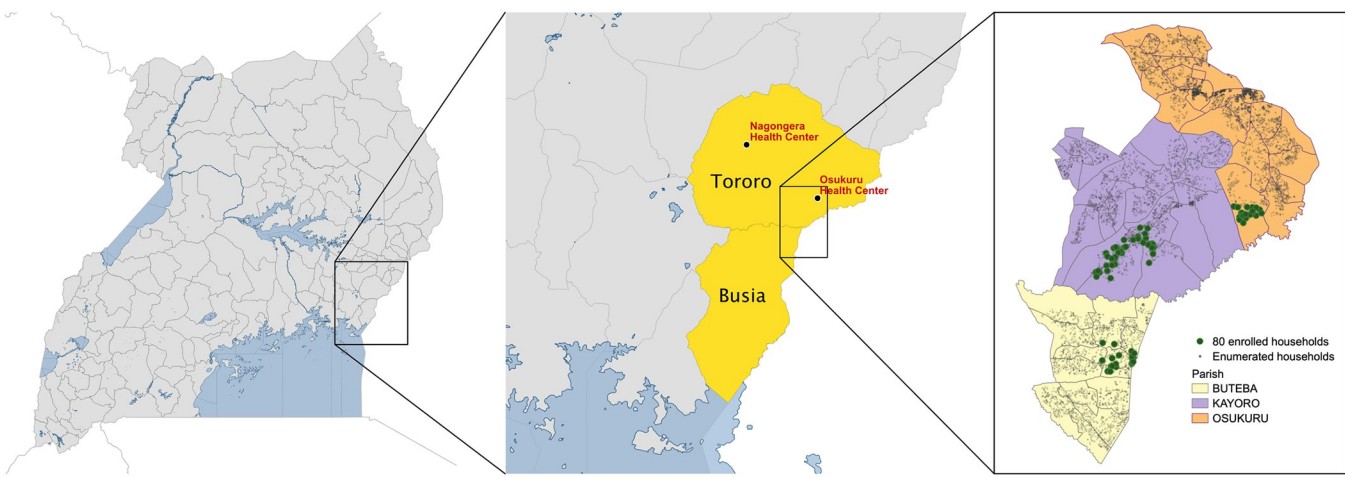

**Fig 1. Map of Uganda with an extract showing study sites.**

pregnancy, and malaria case management with artemether-lumefantrine. In November 2013, universal distribution of free LLINs was conducted as part of a national campaign, and similar campaigns were repeated in May 2017 and June 2020. Indoor residual spraying with the carbamate bendiocarb was initiated in December 2014–January 2015, with additional rounds administered in June–July 2015 and November–December 2015. In June–July 2016, the IRS formulation was changed to the organophosphate pirimiphos-methyl (Actellic), with repeated rounds in June–July 2017, June–July 2018, and March–April 2019. In March-April 2020, the formulation of IRS was changed to Fludora Fusion, a recently prequalified insecticide containing a mixture of clothianidin and deltamethrin [6]. IRS has also been sustained since December 2014 in Butaleja and Bugiri Districts, which border the northern side of Tororo District, but has not been implemented in other districts bordering Tororo or on the Kenyan side of the border with Tororo.

Busia District is a rural district bordering Tororo district to the south. Although no information on EIR is available for Busia District, data from health facility-based surveillance have documented very high-test positivity rates (TPRs) among patients presenting to outpatient facilities in this area [7]. As in Tororo, prior to 2013 malaria control in Busia District was limited to targeted distribution of LLINs through antenatal care services, promotion of intermittent preventive treatment during pregnancy, and malaria case management with AL. Universal distribution of free LLINs was conducted as part of repeated national campaigns in Busia District in 2013, 2017, and in December 2020. IRS has never been implemented in Busia District.

### Routine health-facility based malaria surveillance data

Weekly aggregate data on the number of patients tested for malaria and the number of patients testing positive for malaria were collected retrospectively from the outpatient departments of Nagongera Health Centre IV and Osukuru Health Centre III. Temporal trends in malaria morbidity were generated by calculating the weekly test positivity rate, defined as the number of patients testing positive for malaria / the number of patients tested.

### Screening and enrollment of cohort households

In May 2020, all 10,474 households in the parishes of Osukuru, Kayoro, and Buteba were enumerated and mapped using handheld global positioning systems (Garmin e-Trex 10 GPS unit,

Garmin International Inc., Olathe, KS) to provide a sampling frame for recruitment of households into the cohort study (Fig 1). A household was defined as any single permanent or semi-permanent dwelling structure acting as the primary residence for a person or group of people that generally cook and eat together. Following enumeration, a cross-sectional survey was conducted in 300 randomly selected households to generate spatial estimates of parasite prevalence within the study area. Using the survey results, the study area was stratified into high, medium, and low transmission areas. In August 2020, households in the three transmission areas were randomly selected and screened for eligibility to join the cohort study. Households were enrolled in the cohort if they met the following criteria: 1) having at least two members aged 5 years or younger; 2) no more than 7 permanent residents currently residing; 3) no plans to move from the study catchment area in the next 2 years; and 4) willingness to participate in entomological surveillance studies.

## Screening, enrollment, and follow-up of cohort participants

All permanent residents from enrolled households were screened and enrolled in the cohort study if they met the following criteria: 1) the selected household was considered their primary residence; 2) agreement to come to the study clinic for any febrile illness and scheduled routine visits; 3) agreement to avoid antimalarial medications outside the study; and 4) provision of written informed consent (for parent or guardian in the case of children). The cohort was dynamic, such that over the course of the study any permanent residents that joined a household were screened for enrollment.

At enrollment, a baseline evaluation was conducted which included a detailed medical history, focused physical examination, and blood collection by venipuncture for hemoglobin measurement, thick blood smear, and storage for future molecular studies. A household survey was conducted to collect information on characteristics of the household and LLIN ownership. A wealth index was generated for each household using principal components analysis based on common assets and categorized into tertiles. Cohort study participants were encouraged to come to a dedicated study clinic open 7 days per week for all their medical care. Routine visits were conducted every 4 weeks and included a standardized evaluation and collection of blood by finger prick/heel stick (if < 6 months of age) or venipuncture (if aged 6 months and older) for thick blood smear, hemoglobin measurement (every 12 weeks), and storage for future molecular studies. Study participants found to have a fever (tympanic temperature > 38.0˚C) or history of fever in the previous 24 hours at the time of any clinic visit had a thick blood smear read immediately. If the thick blood smear was positive by light microscopy, the patient was diagnosed with malaria and managed according to national guidelines [8]. Study participants who missed their scheduled routine visits were visited at home and requested to come to the study clinic as soon as possible.

All enrolled participants were followed through January 10th, 2021 unless they were prematurely withdrawn. Participants were withdrawn if they: 1) moved out of the cohort household; 2) were unable to be located for > 4 months; 3) withdrew informed consent; or 4) were unable to comply with the study schedule and procedures.

## Entomological surveillance

Mosquito collections were conducted every 2 weeks in all cohort study households. In each room where study participants slept, a miniature CDC light trap (Model 512; John W. Hock Company, Gainesville, Florida, USA) was positioned 1 m above the floor. Traps were set at 7 PM and collected at 7 AM the following morning. Female *Anopheles* were identified taxonomically to species level based on morphological criteria according to established taxonomic keys

[9]. Every 2 weeks, up to 30 mosquitoes were randomly selected for PCR analysis to identify members of the *Anopheles gambiae* complex [10]. All female *Anopheles* mosquitoes were stored in dessicant and assessed for sporozoites using a standardized ELISA technique [11].

## Laboratory evaluations

Thick blood smears were stained with 2% Giemsa for 30 minutes and evaluated for the presence of asexual and sexual (gametocytes) parasites. Parasite and gametocyte densities were calculated by counting the number of asexual parasites per 200 leukocytes (or per 500, if the count was less than 10 parasites per 200 leukocytes), assuming a leukocyte count of 8,000/μl. A thick blood smear was considered negative if examination of 100 high power fields revealed no asexual parasites. For quality control, all slides were read by a second microscopist, and a third reviewer settled any discrepant readings. Quantitative PCR (qPCR) was performed at the time of enrollment, at each routine visit (every 4 weeks), and when malaria was diagnosed.

At each of these time points, DNA was extracted from approximately 200 μL of whole blood using Qiagen spin columns, and extraction products were tested for the presence and quantity of *P. falciparum* DNA using a highly sensitive qPCR assay targeting the multicopy conserved var gene acidic terminal sequence with a lower limit of detection of 50 parasite/mL [12].

Following automated extraction (Total Nucleic Acid Isolation Kit-High Performance; Roche Applied Science, Indianapolis, IN, USA), quantitative reverse transcriptase PCR (RT-qPCR) targeting male (PfMGET) and female (CCp4) mRNA transcripts was used to quantify gametocytes using 100 μL whole blood in RNA preservative (RNA protect Cell Reagent; Qiagen, Hilden, Germany) with a lower limit of detection of 0.01 gametocytes/ μL [13]. For high parasitemia samples (>1000 parasites/ μL), gametocyte densities were adjusted for background transcripts in asexual parasites [13]. Of all qPCR positive samples (n = 1912), the first qPCR positive follow-up sample from each participant (n = 387) was used for gametocyte quantification. Due to 6 extraction failures, 4 of the first qPCR positive follow-up visits were replaced by the second qPCR positive follow-up visit. This resulted in a total of 385 gametocyte measurements.

## Ethics approval and consent to participate

Ethical approval was obtained from the Makerere University School of Medicine Research and Ethics Committee **(REF 2019–134)**, the Uganda National Council of Science and Technology **(HS 2700)**, the London School of Hygiene & Tropical Medicine Ethics Committee **(17777)**, and the University of California, San Francisco Committee on Human Research **(257790)**. Written informed consent was obtained for all participants prior to enrolment into the study.

## Statistical analysis

All data were collected using standardized case record forms and double-entered using Microsoft Access (Microsoft Corporation, Redmond, Washington, USA). Analyses were performed using Stata, version 14 (Stata Corporation, College Station, Texas, USA) and R version 4.1.0 [14]. Baseline descriptive statistics included proportions for categorical variables and mean (SD) or median (range) values for continuous variables. Measures of transmission were based on entomological surveillance data. The human biting rate (HBR) was estimated using the total number of female *Anopheles* mosquitoes captured / number of CDC light trap collections. The sporozoite rate was calculated as the number of mosquitoes testing positive for sporozoites / the number of mosquitoes tested. The annual EIR was estimated using the product of the daily HBR and the sporozoite rate x 365 days/year.

Clinical metrics, including measures of infection and disease, were estimated and stratified by parish of residence and age categories. The prevalence of microscopic parasitemia/gametocytemia was calculated as the number of routine blood smears positive for asexual parasites or sexual parasites / total number of routine blood smears done. The prevalence of microscopic or sub-microscopic parasitemia was calculated as the number of routine samples positive by microscopy or qPCR / total number of routine assessments. The prevalence of gametocytes by RT-qPCR was calculated as the percentage of gametocyte positive samples among individuals at the time of their first follow-up visit when parasites were detected by qPCR. The incidence of malaria was calculated as the number of incident episodes of symptomatic malaria (defined as fever and a positive thick blood smear not proceeded by another episode of malaria in the prior 14 days) / person years of observation. To assess for measures of clinical immunity to malaria, individual observation time was divided into 28-day intervals (corresponding to the timing of routine clinic visits), and the interval risks of any symptomatic malaria, microscopic parasitemia, and symptomatic malaria if microscopic parasitemia was present were measured.

Pairwise comparisons between the parishes of entomological measures of transmission (HBR, sporozoite rate, and EIR) at the household level were made using Poisson regression models. Pairwise comparisons between the parishes of measures of infection (parasite and gametocyte prevalence using both microscopy and PCR) and interval risks of clinical outcomes were made using mixed effects generalized linear models with a Poisson family and adjustment for repeated measures from the same individual. Pairwise comparisons of malaria incidence between the parishes were made using negative binomial regression models. Assuming a p-value of $< 0.05$ (two-sided) was considered statistically significant.

## Results

### Test positivity rates at health facilities

To provide context for findings from the cohort study presented below, trends in malaria TPRs over a 6.5-year period following the implementation of IRS were compared between Nagongera Health Center, located in the interior of Tororo District, and Osukuru Health Center, located near the eastern border of Tororo District, where the cohort study was conducted. At both health centers, a gradual decline in TPR was recorded from 2015–2018, with peaks proceeding each round of IRS (Fig 2); in Osukuru mean weekly TPR decreased from 45.5% in 2015 to 26.8% in 2018, and in Nagongera mean weekly TPR decreased from 15.4% in 2015 to 6.5% in 2018. At both facilities, TPR remained relatively stable in 2019 and most of 2020, and then began to rise in late 2020; in Osukuru mean weekly TPR reached 39.1% in the first half of 2021, in Nagongera mean weekly TPR reached 24.4% in the first half of 2021. Throughout the 6.5-year period following the implementation of IRS in Tororo, TPR was consistently higher in Osukuru (mean weekly TPR = 35.4%) compared to Nagongera (mean weekly TPR = 15.0%).

### Characteristics of the households and cohort study participants

A total of 326 households were screened and 80 enrolled in the study (20 from Buteba Parish, 40 from Kayoro Parish, and 20 from Osukuru Parish). The primary reason for exclusion was having fewer than 2 household residents under 5 years of age (Fig 3). Characteristics of the study households and cohort participants are presented in Table 1. Houses from Osukuru were more likely to be constructed using traditional materials and houses from Buteba were more likely to be in the poorest wealth category. As expected, no households in Buteba, 90% of households in Kayoro, and all households in Osukuru reported receiving IRS in the last 12 months.

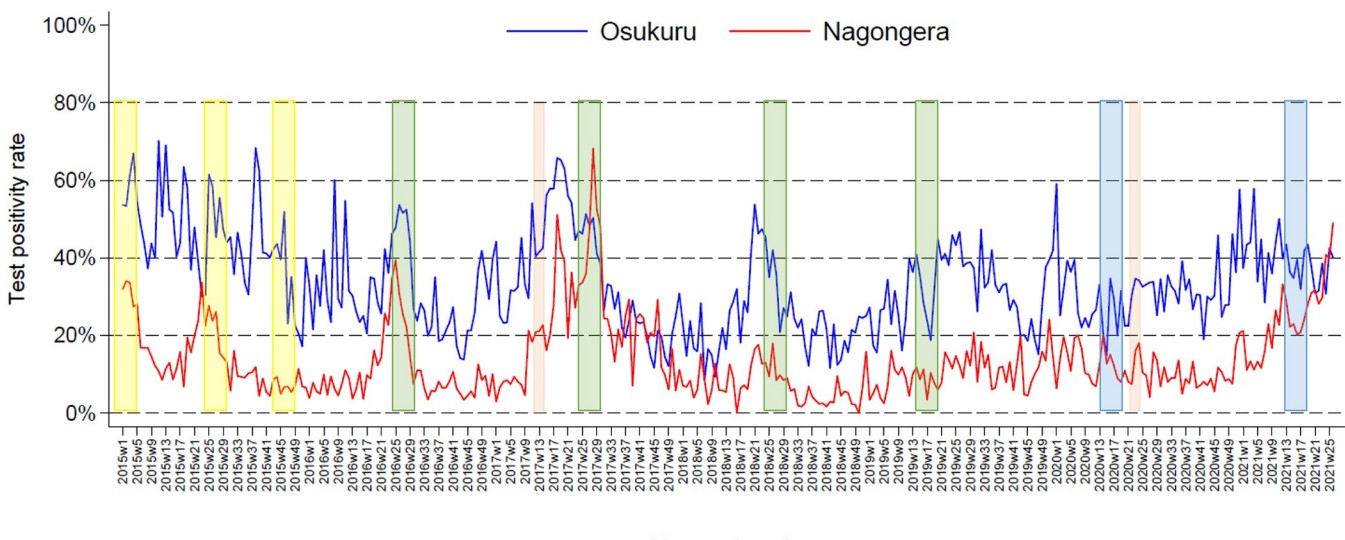

**Fig 2. Temporal changes in test positivity rates estimated from routine health facility-based data (January 2015 –June 2021) from the interior of Tororo District (Nagongera Health Centre IV) and within the border area of Tororo District where the cohort study was conducted (Osukuru Health Centre III).** (Pink bars signify rounds of mass LLIN distribution, yellow bars are rounds of IRS with bendiocarb, green bars are rounds of IRS with pirimiphos-methyl, and blue bars are rounds of IRS with Fludora Fusion).

A mass campaign to distribute LLINs was conducted 3 months prior to cohort enrollment in Tororo district, and 3 months after enrollment in Busia. At the time of enrollment, the proportion of households that reported owning at least one LLIN was 45% in Kayoro and 65% in Osukuru (Tororo district), and only 20% in Buteba (Busia district). All cohort households were provided LLINs by the study team at the time of enrollment. All 483 household residents were enrolled in the cohort study (119 from Buteba, 247 from Kayoro, and 117 from Osukuru). Of the enrolled participants, 262 (54.2%) were female, and the median age at enrollment was 8.6 years (interquartile range 3.5–25.7), with a similar distribution of age and gender across the three parishes (Table 1).

## Measures of transmission

Entomological measures of transmission stratified by parish are presented in Table 2. Overall, 19,563 female *Anopheles* mosquitoes were collected in 1,190 overnight CDC light trap collections (Table 2). *Anopheles* species composition differed between the two districts. In Buteba, where IRS was not implemented, the majority (85.4%) of mosquitoes collected were *Anopheles gambiae s.s.* In Kayoro and Osukuru, where IRS has been sustained for an extended period, less than 50% of mosquitoes were *Anopheles gambiae s.s.*, with *Anopheles arabiensis* and *Anopheles funestus* much more common than in Buteba. The daily HBR was significantly higher in Buteba (23.6) compared to Kayoro (13.1 p<0.001), and Osukuru (17.4, p<0.001). The annual EIR was significantly higher in Buteba (108.2 infective bites/person/year) compared to Osukuru (59.0, p = 0.002) and Kayoro (27.4, p<0.001).

## Measures of infection and disease

Routine visits were conducted at enrollment and then every 4 weeks, resulting in 2,523 scheduled visits over the study period (Table 3). Overall, parasite prevalence was 19.5% by microscopy and 50.7% by qPCR, with no significant differences across the three parishes. When

**326 households screened**

108 Buteba Parish
140 Kayoro Parish
78 Osukuru Parish

**246 households excluded**

211 less that 2 household residents under 5 years of age
16 houses not occupied
10 houses with more than 7 residents
5 not all residents agreed to screening
4 refused entomology studies

**80 households enrolled**

20 Buteba Parish
40 Kayoro Parish
20 Osukuru Parish

**450 household members enrolled during initial screening**

101 Buteba Parish
239 Kayoro Parish
110 Osukuru Parish

**33 additional household members enrolled during dynamic screening**

18 Buteba Parish
8 Kayoro Parish
7 Osukuru Parish

**483 total household members enrolled**

119 Buteba Parish
247 Kayoro Parish
117 Osukuru Parish

**20 household members withdrawn**

16 lost to follow-up
4 moved out of study household

**463 household members completed follow-up**

104 Buteba Parish
242 Kayoro Parish
117 Osukuru Parish

**Fig 3. Flow diagram of the cohort study.**

**Table 1. Characteristics of households and cohort participants at enrolment.**

| Characteristic | | Busia District | Tororo District | | |
| --- | --- | --- | --- | --- | --- |
| | | Buteba Parish | Kayoro Parish | Osukuru Parish | |
| **Household characteristics** | | | | | |
| Number of Households | | 20 | 40 | 20 | |
| Residents per household, median (range) | | 4.5 (3–7) | 6 (4–7) | 6 (4–7) | |
| Type of housing construction, n (%) | Traditional | 12 (60.0) | 27 (67.5) | 20 (100) | |
| | Modern | 8 (40.0) | 13 (32.5) | 0 | |
| Wealth category, n (%) | Poorest | 14 (70.0) | 11 (27.5) | 8 (40.0) | |
| | Middle | 3 (15.0) | 12 (30.0) | 7 (35.0) | |
| | Least poor | 3 (15.0) | 17 (42.5) | 5 (25.0) | |
| Number of rooms used for sleeping, median (range) | | 1 (1–3) | 2 (1–4) | 1 (1–3) | |
| Number of sleeping spaces, median (range) | | 2 (2–6) | 3 (2–5) | 3 (2–5) | |
| IRS in the last 12 months, n (%) | | 0 | 36 (90.0) | 20 (100) | |
| Households with at least 1 LLIN, n (%) | | 4 (20.0) | 18 (45.0) | 13 (65.0) | |
| Households with 1 LLIN per 2 persons, n (%) | | 2 (10.0) | 4 (10.0) | 4 (20.0) | |
| **Participant characteristics** | | | | | |
| Number of participants | | 119 | 247 | 117 | |
| Female gender, n (%) | | 65 (54.6) | 132 (53.4) | 65 (55.6) | |
| Age in years, median (range) | | 8.2 (0.2–64) | 8.7 (0.05–68) | 9.1 (0.12–59) | |
| Age categories, n (%) | < 5 | 49 (41.2) | 92 (37.2) | 42 (35.9) | |
| | 5–15 years | 29 (24.4) | 72 (29.2) | 36 (30.8) | |
| | > 15 years | 41 (34.4) | 83 (33.6) | 39 (33.3) | |

stratified by age, patterns of parasite prevalence by microscopy and qPCR were similar across the 3 parishes (Fig 4). For all parishes combined, parasite prevalence by microscopy was higher among those 5–15 years of age (34.7%) compared to those < 5 years (14.6%, p<0.001) and those over 15 years (12.3%, p<0.001), with no significant difference between those < 5 years and those > 15 years (p = 0.29). Parasite prevalence by qPCR was also higher among those 5–15 years (66.3%) compared to those < 5 years (39.3%, p<0.001) and those over 15 years (50.3%, p<0.001), but also higher in those over 15 years compared to those < 5 years (p = 0.002). Overall, the prevalence of gametocytes was 2.0% by microscopy and 45.5% by RT-

**Table 2. Entomological metrics from bi-weekly CDC light trap collections stratified by parish.**

| Metric | | Busia District | Tororo District | |
| --- | --- | --- | --- | --- |
| | | Buteba Parish | Kayoro Parish | Osukuru Parish |
| Total collections | | 261 | 645 | 284 |
| Total female *anopheles* collected | | 6170 | 8447 | 4946 |
| Relative proportion of different *Anopheles* species | *Anopheles gambiae s.s.* | 85.4% | 48.4% | 29.2% |
| | *Anopheles arabiensis* | 8.4% | 37.3% | 32.9% |
| | *Anopheles funestus* | 5.5% | 12.2% | 34.0% |
| | Other *Anopheles* species | 0.7% | 2.0% | 3.8% |
| Daily human biting rate | | 23.6 | 13.1 | 17.4 |
| Tested for sporozoites | | 4546 | 8031 | 4526 |
| Positive for sporozoites | | 57 | 46 | 42 |
| Sporozoite rate | | 1.25% | 0.57% | 0.93% |
| Annual entomological inoculation rate | | 108.2 | 27.4 | 59.0 |

Table 3. Measures of infection and disease stratified by parish.

| Metric | Busia District | Tororo District | |
|---|---|---|---|
| | Buteba Parish | Kayoro Parish | Osukuru Parish |
| **Measures of infection** | | | |
| Total scheduled visits | 645 | 1199 | 679 |
| Parasite prevalence by microscopy, n (%) | 143 (22.2) | 216 (18.0) | 134 (19.7) |
| Parasite density by microscopy*, geometric mean | 652 | 501 | 581 |
| Parasite prevalence by qPCR, n (%) | 338 (52.4) | 571 (47.6) | 371 (54.6) |
| Parasite density by qPCR*, geometric mean | 2.7 | 1.7 | 1.9 |
| Gametocyte prevalence by microscopy, n (%) | 9 (1.4) | 26 (2.2) | 15 (2.2) |
| Gametocyte prevalence by RT-qPCR, n (%)** | 42/93 (45.2) | 89/183 (48.6) | 44/109 (40.4) |
| Gametocyte density by RT-qPCR*, geometric mean | 0.35 | 0.81 | 0.50 |
| **Measures of disease** | | | |
| Person years of follow up | 43.4 | 89.3 | 48.3 |
| Incident episodes of malaria | 64 | 97 | 119 |
| Parasite density by microscopy*, geometric mean | 5591 | 5032 | 4925 |
| Incidence of malaria per person years | 1.47 | 1.09 | 2.46 |

* Parasites / μL among those with parasites detected

** Considering only the first sample following enrolment that was positive for any parasites by qPCR.

qPCR among those with parasites detected by qPCR, with no significant differences across the three parishes.

A total of 280 episodes of symptomatic malaria were diagnosed over 181 person years of follow-up (Table 3); 277 episodes were uncomplicated and treated with AL and 3 episodes occurred in children with danger signs who were treated with IV artesunate, followed by oral AL (2 from Buteba and 1 from Kayoro). No cases met criteria for severe malaria. The incidence of malaria was significantly higher in Osukuru (2.46 episodes/person/per year) compared to Buteba (1.47, p = 0.005) and Kayoro (1.09, p<0.001). When comparing the incidence of malaria across parishes stratified by age, the incidence of malaria was more than twice as high in Oskuru compared to Buteba and Kayoro for children < 5 years of age and 5–15 years of age (Fig 4). As expected, the incidence of malaria was lower in participants aged 15 years or older compared to what was observed in children and there were no significant differences in estimates between the 3 parishes for those aged 15 years of older. When comparing Buteba to Kayoro, the incidence of malaria was significantly different only among children < 5 years (2.28 vs. 1.41, p = 0.03).

To assess for differences in clinical immunity, interval risks of symptomatic malaria, microscopic parasitemia, and the joint probability of symptomatic malaria if microscopic parasitemia present were compared between the parishes after stratification by age groups (Table 4). Using Buteba (the parish with the highest transmission intensity) as the reference group, the relative risks for symptomatic malaria increased with increasing age for both Kayoro and Osukuru, although pairwise comparisons were only statistically significant for Osukuru among the < 5 year and 5–15 year age groups. There were no significant differences in the interval risk of microscopic parasitemia between Buteba and Kayoro or Osukuru, although in Osukuru relative risks decreased with increasing age, the opposite of the trend seen with symptomatic malaria. For the combined risk of symptomatic malaria if microscopic parasitemia present (an indicator of less clinical immunity), compared to Buteba the relative risks increased with

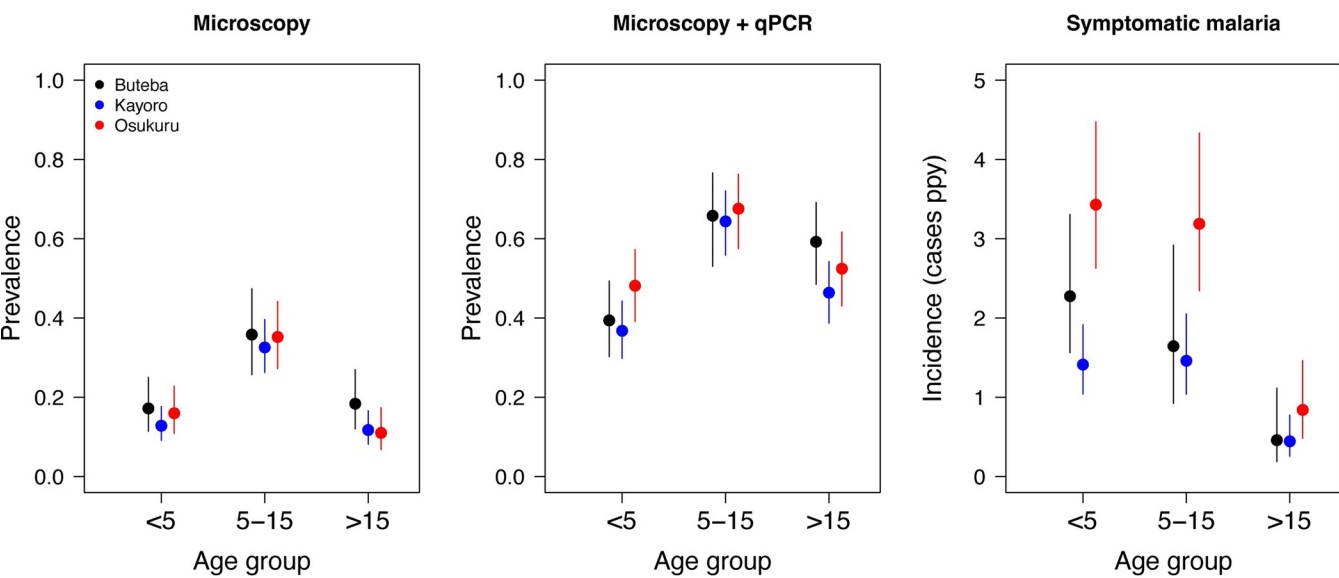

**Fig 4. Prevalence of microscopic parasitemia and microscopic+ submicroscopic parasitemia as well as incidence of malaria across the three parishes stratified by age groups.** All estimates (points) and 95% confidence intervals (lines) were calculated using generalized estimating equations with a binomial family and exchangeable correlation structure to account for repeated measures within individuals.

increasing age for both Kayoro and Osukuru, with those over 15 years of age in Osukuru being almost 3 times more likely to have symptomatic malaria if microscopic parasitemia present compared to Buteba (RR = 2.99, 95% CI 1.15–7.78, p = 0.03) (Table 4).

## Discussion

Like many other countries in sub-Saharan Africa, malaria prevention in Uganda has largely relied on vector control through large scale LLIN distribution campaigns and IRS in selected areas. This approach has been highly successful, with an 84% reduction in malaria cases documented in five districts of Uganda 4–5 years after IRS was first implemented in 2014. However,

**Table 4. Associations between parish and clinical outcomes stratified by age.**

| Age strata | District | Parish | Intervals* | Interval risk of any symptomatic malaria | | | Interval risk of any microscopic parasitemia | | | Risk of symptomatic malaria if microscopic parasitemia present** | | |
|---|---|---|---|---|---|---|---|---|---|---|---|---|
| | | | | n (%) | RR (95% CI) | p-value | n (%) | RR (95% CI) | p-value | % | RR (95% CI) | p-value |
| < 5 years | Busia | Buteba | 260 | 39 (15.0) | Reference | | 63 (24.2) | Reference | | 61.9 | Reference | |
| | Tororo | Kayoro | 439 | 45 (10.3) | 0.68 (0.44–1.06) | 0.09 | 85 (19.4) | 0.81 (0.55–1.19) | 0.28 | 52.9 | 0.86 (0.56–1.31) | 0.48 |
| | | Osukuru | 247 | 59 (23.9) | 1.60 (1.05–2.42) | 0.03 | 81 (32.8) | 1.41 (0.94–2.12) | 0.10 | 72.8 | 1.18 (0.79–1.76) | 0.43 |
| 5–15 years | Busia | Buteba | 162 | 18 (11.1) | Reference | | 70 (43.2) | Reference | | 25.7 | Reference | |
| | Tororo | Kayoro | 354 | 40 (11.3) | 1.01 (0.57–1.82) | 0.96 | 138 (39.0) | 0.90 (0.67–1.21) | 0.49 | 29.0 | 1.13 (0.64–2.00) | 0.68 |
| | | Osukuru | 204 | 41 (20.1) | 1.81 (1.01–3.27) | 0.047 | 92 (45.1) | 1.04 (0.76–1.43) | 0.79 | 44.6 | 1.74 (0.98–3.09) | 0.06 |
| > 15 years | Busia | Buteba | 225 | 6 (2.7) | Reference | | 41 (18.2) | Reference | | 14.3 | Reference | |
| | Tororo | Kayoro | 407 | 11 (2.7) | 1.07 (0.36–3.18) | 0.91 | 49 (12.0) | 0.64 (0.38–1.08) | 0.10 | 22.5 | 1.53 (0.57–4.15) | 0.40 |
| | | Osukuru | 228 | 14 (6.1) | 2.38 (0.80–7.08) | 0.12 | 32 (14.0) | 0.78 (0.43–1.39) | 0.39 | 43.8 | 2.99 (1.15–7.78) | 0.03 |

* approximately 28-day periods of observation between routine visits

** Number of intervals with any symptomatic malaria diagnosed / number of intervals with any microscopic parasitemia detected

more recently malaria cases have markedly increased despite sustained IRS, exceeding pre-IRS levels by 39% in these same five districts [5]. Although causes of this resurgence are not clear, the timing corresponded to a change in active ingredient used for IRS. This study compared the epidemiology of malaria in a contiguous area that included two parishes in Tororo District early in the resurgence despite sustained IRS with one parish in Busia District where LLINs were IRS has never been implemented. Entomologic measures of transmission intensity were modestly lower in the two parishes from Tororo District compared to the parish in Busia District. In contrast, parasite and gametocyte prevalence were similar across the three parishes and malaria incidence was highest in one of the parishes in Tororo, where there was evidence of waning immunity against symptomatic malaria in older individuals.

Confirming recent observations from routine health facility data [5], the burden of malaria in the studied parishes of Tororo were much higher compared to those measured a few years earlier in Nagongera sub-county, located in the interior of Tororo District. Indeed, using the same methodology as this study, malaria metrics in a cohort of all ages in Nagongera had reached pre-elimination levels after 5 years of sustained IRS, with an annual EIR of < 1 infectious bite/person/year, parasite prevalence by microscopy < 2%, and an incidence of malaria of 0.04 episodes/person/year [2].

There are a several potential explanations as to why the burden of malaria was much higher a few years later in the two parishes from Tororo District included in this study that bordered areas were IRS has not been implemented. First, repeated rounds of IRS may not be as effective or have a shorter duration of effect in areas that border high transmission areas that have not received as intensive vector control measures. This is supported by the health facility-based data included in this study, which consistently showed higher TPRs from the facility located near the border where IRS has not been implemented compared to the facility located in the interior of Tororo District. Movement of people and mosquitos infected with malaria parasites across borders has been well documented as one of the major challenges to malaria control and elimination efforts especially between countries [15, 16]. Indeed, there are several examples from Southern Africa where IRS has been highly effective in reducing the burden of malaria, however, achieving elimination has been challenged by the importation of malaria across borders [17].

Second, is the possibility that the effectiveness of IRS in Tororo District may have declined recently, corresponding to the interval period from when the previous study in Nagongera sub-county and this study were conducted. This is also supported by the heath facility-based data included in this study which showed an increase in TPR since late 2019 at both facilities. The dramatic decline in malaria burden documented in Nagongera sub-county after 5 years of sustained IRS involved round of the carbamate bendiocarb every 6 months (2014–2015) followed by annual rounds with the organophosphate Actellic (2016–2019) [2]. However, in March 2020 there was a switch to Fludora Fusion, a novel insecticide combination containing clothianidin and deltamethrin that was recently added to the WHO's list of pre-qualified insecticides for use in indoor residual spraying [6]. Fludora Fusion has been shown to be effective in reducing measures of transmission intensity, especially in areas where pyrethroid-resistance has been reported [18, 19]. However, data on Fludora Fusion have been limited to laboratory-based or experimental hut studies and none have evaluated clinically relevant outcomes in "real world" settings. In addition, the recent effectiveness of the IRS program may have been compromised by problems with implementation, especially in the context of the recent COVID-19 pandemic. Although, households included in this study reported high coverage levels during the last round of IRS, other problems may have occurred such as spray operators spending less time in homes due to fear of acquiring COVID. Indeed, disruptions of essential

malaria services, including IRS, may have played a role in a surge in malaria cases documented during in the first half of 2020 in Zimbabwe [20].

Third, species composition and behavioral characteristics of mosquitoes may have played a role in the relatively high burden of malaria in the Kayoro and Osukuru parishes. Prior to the implementation of IRS, the predominant species collected in Nangongera was *Anopheles gambiae s.s.*. However, after 5 years of sustained IRS, there was a dramatic shift in species composition, with *Anopheles arabiensis* making up over 98% of mosquitoes collected [2, 3]. Compared to Nagongera, species composition was quite different in Kayoro and Osukuru parishes, with *Anopheles arabiensis* making up only a third of mosquitoes collected, and the remainder primarily *Anopheles gambiae s.s or Anopheles funestes.* Another contributing factor could have been a shift from indoor to outdoor biting, circumventing IRS targeting indoor biting mosquitoes, which has been reported in Tororo District [21].

Another interesting observation from this study was that, compared to Buteba parish in Busia district, where IRS has never been implemented, Osukuru parish in Tororo district had a higher incidence of malaria, despite lower transmission intensity. In addition, older parasitemic individuals in Osukuru were more likely to develop symptomatic malaria, suggesting less clinical immunity. It is well known that, in high transmission areas, older children and adults gradually develop immunity against malaria infection and disease, although this immunity is incomplete and not permanent [22]. One possible explanation for the findings of this study is that immunity waned among the population of Tororo District during the years when IRS was highly effective, evidenced by the more modest decrease in risk of developing symptomatic malaria in the setting of microscopic parasitemia seen with increasing age at these sites. If relatively high transmission intensity in Osukuru parish was a recent phenomenon, due to loss of IRS efficacy or other factors, this could have resulted in an unusually high burden of symptomatic malaria in a population that had become relatively nonimmune. An analogous scenario has been described in an area of northern Uganda, where the withdrawal of IRS after an extended period was associated with a remarkably high malaria burden, even exceeding pre-IRS levels in patients over 5 years of age [23].

This study had several limitations. First, data from the three study areas represented only a "snapshot" in time, and care should be taken when comparing data collected previously in a different area of Tororo District. Second, entomological measures of transmission intensity were generated using CDC light traps and therefore only surrogate measures of human biting behavior that did not capture outdoor biting mosquitoes. Third, in addition to impact of IRS, which was the primary focus of this study, differences in malaria metrics between the 3 parishes could have been due to a number of other factors that we were unable to measure. Fourth, given the labor-intensive nature of this comprehensive cohort study, the sample size was relatively small, limiting statistical power to assess associations and make precise estimates generalizable to the larger population.

## Conclusion

In a border area where IRS has been sustained for 6 years, measures of malaria transmission, infection and disease were comparable to those in an adjacent area where IRS had not been implemented and much higher than recent estimates from the interior of the district receiving IRS. These observations highlight the importance of detailed surveillance studies to better understand the scope and durability of malaria control interventions targeting specific geographic areas. Further studies are warranted to continue to monitor the effectiveness of IRS in Uganda, the causes of the recent decline in effectiveness, and implications for sustaining gains and consideration of IRS exit strategies.

## Availability of data and materials

Data from both cohort studies are available through a novel open-access clinical epidemiology database resource, ClinEpiDB [24].

## Acknowledgments

We thank the study team and the Infectious Diseases Research Collaboration (IDRC) for administrative and technical support. We are grateful to the study participants who participated in this study and their families.

## Author Contributions

**Conceptualization:** Joaniter I. Nankabirwa, Emmanuel Arinaitwe, Bryan Greenhouse, Moses R. Kamya, Grant Dorsey.

**Data curation:** Teun Bousema, Sara Lynn Blanken, John Rek, Sarah G. Staedke.

**Formal analysis:** Joaniter I. Nankabirwa, Teun Bousema, Sara Lynn Blanken, Grant Dorsey.

**Funding acquisition:** Joaniter I. Nankabirwa, Grant Dorsey.

**Methodology:** Joaniter I. Nankabirwa, John Rek, Bryan Greenhouse, Philip J. Rosenthal, Moses R. Kamya, Grant Dorsey.

**Supervision:** Joaniter I. Nankabirwa, Emmanuel Arinaitwe, Moses R. Kamya, Grant Dorsey.

**Writing – original draft:** Joaniter I. Nankabirwa, Grant Dorsey.

**Writing – review & editing:** Joaniter I. Nankabirwa, Teun Bousema, Sara Lynn Blanken, John Rek, Emmanuel Arinaitwe, Bryan Greenhouse, Philip J. Rosenthal, Moses R. Kamya, Sarah G. Staedke, Grant Dorsey.

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
