## [Decision Letter · Decision Letter 0]

2 Nov 2022

PONE-D-22-27386Measures of malaria transmission, infection, and disease in an area bordering two districts with and without sustained indoor residual spraying of insecticide in UgandaPLOS ONE

Dear Dr. Nankabirwa,

Thank you for submitting your manuscript to PLOS ONE. After careful consideration, we feel that it has merit but does not fully meet PLOS ONE’s publication criteria as it currently stands. Therefore, we invite you to submit a revised version of the manuscript that addresses the points raised during the review process.

We look forward to receiving your revised manuscript.

Kind regards,

Luzia H Carvalho, Ph.D.

Academic Editor

PLOS ONE

Journal Requirements:

Additional Editor Comments:

After careful consideration, we felt that your manuscript requires substantial revision, following which it can possibly be reconsidered, thus governing the decision of a “major revision”. According to the reviewer #2, it is not clear whether the current MS is hypothesis-driven research. Consequently, the Authors should clarify the overall purpose of the MS. At this point, the reviewer suggests that additional data may be necessary to address some statements raised by the authors (epidemiological, entomological and environmental data).  A significant number of issues raised by both reviewers should be clarified and/or adjust. Finally, it is essential to incorporate the limitations of the study otherwise it might compromise data interpretation.   For your guidance, a copy of the reviewers' comments was included below.

Reviewers' comments:

Reviewer's Responses to Questions

**Comments to the Author**

1. Is the manuscript technically sound, and do the data support the conclusions?

Reviewer #1: Yes

Reviewer #2: Partly

2. Has the statistical analysis been performed appropriately and rigorously? 

Reviewer #1: Yes

Reviewer #2: Yes

3. Have the authors made all data underlying the findings in their manuscript fully available?

Reviewer #1: Yes

Reviewer #2: Yes

4. Is the manuscript presented in an intelligible fashion and written in standard English?

Reviewer #1: Yes

Reviewer #2: Yes

5. Review Comments to the Author

Reviewer #1: Great piece of work and written in a very readable way, however, there are few occasions were mosquito species were written inappropriately. For instance, line 173 and Table 2 but also some minor grammatical errors such as in the second line (i.e. IRS is) of the first paragraph of the discussion, figure 3 (i.e. 211 less that) and line 55.

Additionally, the manuscript lack sufficient references throughout, For instance the setup of the CDC light trap has not been backed up at all, etc.

The EIR estimation was correctly done, however, the biting rate should be corrected using coefficient described by Okumu et al, 2010.

Reviewer #2: I thank the author for submitting this manuscript and for the strong work done in its preparation. I think it is well written and a very nice description of the epidemiological situation in western Uganda. My only problem with it, and the reason I have recommended it to be rejected, is that I do not see it as hypothesis-driven research. There is no clearly stated hypothesis in the introduction, with the closest statement that could be considered a goal of the manuscript being "...it is important to better understand the underlying epidemiology of malaria in areas where resurgence has occurred." And while I completely agree with this statement, and feel this manuscript does this very well, I do not think that alone fits the criteria for publication.

The conclusions offer some guidance as to how this study could be more hypothesis-driven, though the authors rightly point out that the manuscript does not answer those outstanding questions as to what the causes of the resurgence are or were. It is possible that there are sufficient data to answer this that were not reported in the manuscript, though the short time frame of the longitudinal data collection, as well as the dual seasonality of malaria in Uganda, make this unlikely. Additional data that I feel would be necessary to answer the above question would be:

- at least one full year of longitudinal data (epidemiological and entomological)

- indoor and outdoor vector biting and infectivity rates (and EIRs)

- species-specific infectivity rates

- IRS residual efficacy data

- mobility data, both between study areas and the broader region

- climate, weather, and environmental data

- other intervention coverage, such as case management indicators if available

The other thought I had when reading the manuscript is that the authors intended it to be focused on the value of using different indicators to measure transmission intensity. If this were in fact the case I once again feel that it would need to be more focused on a specific hypothesis, for example using incidence episodes as opposed to prevalence to show recent changes in transmission intensity. This would also require additional data, likely including different study sites.

Once again I thank the author for the submission and congratulate all authors on the overall quality of the work done, and hope they find these comments useful.

6. PLOS authors have the option to publish the peer review history of their article (what does this mean?). If published, this will include your full peer review and any attached files.

Reviewer #1: No

Reviewer #2: **Yes: **James Colborn, PhD, MSPH

---

## [Author Response · Author response to Decision Letter 0]

17 Nov 2022

Editor’s comments

Comment 1: Please ensure that your manuscript meets PLOS ONE's style requirements.

Response: This has been done

Comment 2: Please include a complete copy of PLOS’ questionnaire on inclusivity in global research in your revised manuscript

Response: A copy of the completed questionnaire has been included in the re-submission. 

Comment 3: Please ensure that you have an ORCID iD and that it is validated in Editorial Manager.

Response: This has been done.

Comment 4: We note that Figure 1 in your submission contain [map/satellite] images which may be copyrighted. All PLOS content is published under the Creative Commons Attribution License (CC BY 4.0), which means that the manuscript, images, and Supporting Information files will be freely available online, and any third party is permitted to access, download, copy, distribute, and use these materials in any way, even commercially, with proper attribution. For these reasons, we cannot publish previously copyrighted maps or satellite images created using proprietary data, such as Google software (Google Maps, Street View, and Earth).

Response: The shapefiles used for figure 1 all came from Uganda Bureau of Statistics and were downloaded through humdata.org. According to their website, the license is Creative Commons Attribution for Intergovernmental Organisations (CC BY-IGO). They are not copywritten and are publicly available for download and use.

Comment 5: Your ethics statement should only appear in the Methods section of your manuscript.

Response: The ethics statement has been moved to the methods section.

Reviewer #1:

Comment 1: Great piece of work and written in a very readable way

Response: Thank you for the comment, we greatly appreciate the encouragement

Comment 2: There are few occasions where mosquito species were written inappropriately. For instance, line 173 and Table 2

Response: These have been corrected to “Anopheles” 

Comment 3: Some minor grammatical errors such as in the second line of the first paragraph of the discussion (i.e. IRS is). 

Response: These errors have been corrected 

Comment 4: Additionally, the manuscript lacks sufficient references throughout, For instance the setup of the CDC light trap has not been backed up at all, etc.

Response: The CDC light trap model used is included in the manuscript as (Model 512; John W. Hock Company, Gainesville, Florida, USA). Because the set up was modified to fit our setting based on previous studies, we have opted to describe the set up. No change has been done in the manuscript. Attempts have been made to include all the literature cited during the writing.

Comment 5: The EIR estimation was correctly done, however, the biting rate should be corrected using coefficient described by Okumu et al, 2010.

Response: We appreciate the recommendations of adjusting for the covariates to the biting rates estimates and thank you for sharing the paper to support the calculations. Unfortunately, we only used CDC light collections and as pointed out in different papers, the collection method is not recommended for the outdoor collections and thus we are missing some of the data/variables needed to make the adjustments in outcome measures. We still believe that the un-adjusted estimates are valid although they may not be as accurate as would be with the adjustments. This was included as a limitation to the entomology study findings (Page 25, lines 445-447).

Reviewer #2:

Comment 1: I thank the author for submitting this manuscript and for the strong work done in its preparation. I think it is well written and a very nice description of the epidemiological situation in western Uganda. 

Response: We appreciate this encouraging comment.

Comment 2: My only problem with it, and the reason I have recommended it to be rejected, is that I do not see it as hypothesis-driven research. There is no clearly stated hypothesis in the introduction, with the closest statement that could be considered a goal of the manuscript being "...it is important to better understand the underlying epidemiology of malaria in areas where resurgence has occurred." And while I completely agree with this statement, and feel this manuscript does this very well, I do not think that alone fits the criteria for publication.

Response: The overall objective of this study was to compare the epidemiology of malaria in an area bordering two regions where indoor residual spraying has never been done and where indoor residual spraying has begun to fail after a sustained period of use. Our underlying hypothesis was that there would be important difference in epidemiological measures of malaria between these two regions that could further our understanding of the epidemiological consequences of resurgent malaria in area where vector control interventions have diminished effectiveness. To address this comment, we have added this statemen to the end of the introduction section.

Comment 3: The conclusions offer some guidance as to how this study could be more hypothesis-driven, though the authors rightly point out that the manuscript does not answer those outstanding questions as to what the causes of the resurgence are or were. It is possible that there are sufficient data to answer this that were not reported in the manuscript, though the short time frame of the longitudinal data collection, as well as the dual seasonality of malaria in Uganda, make this unlikely. Additional data that I feel would be necessary to answer the above question would be:

- at least one full year of longitudinal data (epidemiological and entomological)

- indoor and outdoor vector biting and infectivity rates (and EIRs)

- species-specific infectivity rates

- IRS residual efficacy data

- mobility data, both between study areas and the broader region

- climate, weather, and environmental data

- other intervention coverage, such as case management indicators if available

Response: We appreciate that additional longitudinal data would have made the study stronger, however, for this study, follow-up is limited to the 4 months of follow-up. This was acknowledged as a limitation in the manuscript (Page 25, lines 443-445). Plans are currently underway to conduct additional studies as recommended by the reviewer. 

Comment 4: The other thought I had when reading the manuscript is that the authors intended it to be focused on the value of using different indicators to measure transmission intensity. If this were in fact the case I once again feel that it would need to be more focused on a specific hypothesis, for example using incidence episodes as opposed to prevalence to show recent changes in transmission intensity. This would also require additional data, likely including different study sites.

Response: We agree that it would be interesting to address these questions in different sites with different epidemiological settings. However, in this study we chose to compare our epidemiological parameters of transmission, infection, and disease in 2 districts within a relatively small contiguous area where the only major difference was that one district has never had IRS implemented (and presumable has had relatively stable, high transmission intensity) and the other district was experiencing a recent resurgence in malaria following several years of effective control with IRS. Although we recognize that there are several limitations to our study design and the conclusions we can make, we do feel that the observed differences we describe provide useful information for the public health consequence of malaria resurgence in a historically high endemic African setting. 

Comment 5: Once again I thank the author for the submission and congratulate all authors on the overall quality of the work done, and hope they find these comments useful.

Response: We greatly appreciate the time taken to review the manuscript and the comments provided.

---

## [Decision Letter · Decision Letter 1]

7 Dec 2022

Measures of malaria transmission, infection, and disease in an area bordering two districts with and without sustained indoor residual spraying of insecticide in Uganda

PONE-D-22-27386R1

Dear Dr. Nankabirwa,

We’re pleased to inform you that your manuscript has been judged scientifically suitable for publication and will be formally accepted for publication once it meets all outstanding technical requirements.

Kind regards,

Luzia H Carvalho, Ph.D.

Academic Editor

PLOS ONE

Additional Editor Comments (optional):

Reviewers' comments:

Reviewer's Responses to Questions

**Comments to the Author**

1. If the authors have adequately addressed your comments raised in a previous round of review and you feel that this manuscript is now acceptable for publication, you may indicate that here to bypass the “Comments to the Author” section, enter your conflict of interest statement in the “Confidential to Editor” section, and submit your "Accept" recommendation.

Reviewer #1: All comments have been addressed

Reviewer #2: All comments have been addressed

2. Is the manuscript technically sound, and do the data support the conclusions?

Reviewer #1: Yes

Reviewer #2: (No Response)

3. Has the statistical analysis been performed appropriately and rigorously? 

Reviewer #1: Yes

Reviewer #2: (No Response)

4. Have the authors made all data underlying the findings in their manuscript fully available?

Reviewer #1: Yes

Reviewer #2: (No Response)

5. Is the manuscript presented in an intelligible fashion and written in standard English?

Reviewer #1: Yes

Reviewer #2: (No Response)

6. Review Comments to the Author

Reviewer #1: (No Response)

Reviewer #2: I appreciate the authors' candid responses to my comments, and for taking the time to consider and respond to them. I appreciate the need and desire to publish data and results as they become available rather than waiting for all potential questions to be unequivocally resolved. I look forward to reading any future work that helps to further address the important questions raised in this manuscript.

7. PLOS authors have the option to publish the peer review history of their article (what does this mean?). If published, this will include your full peer review and any attached files.

Reviewer #1: No

Reviewer #2: **Yes: **James Colborn

---

## [Editor Report · Acceptance letter]

21 Dec 2022

PONE-D-22-27386R1 

Measures of malaria transmission, infection, and disease in an area bordering two districts with and without sustained indoor residual spraying of insecticide in Uganda 

Dear Dr. Nankabirwa:

I'm pleased to inform you that your manuscript has been deemed suitable for publication in PLOS ONE. Congratulations! Your manuscript is now with our production department. 

Kind regards, 

on behalf of

Dr. Luzia H Carvalho 

Academic Editor

PLOS ONE